# Protocol for a feasibility randomised controlled trial of targeted oxygen therapy in mechanically ventilated critically ill patients

Daniel S Martin,[1,2] Chris Brew-Graves,[3] Neil McCartan,[3] Gavin Jell,[2] Ingrid Potyka,[3] Jia Stevens,[1,2] Norman R Williams,[3] Margaret McNeil,[1] B Ronan O'Driscoll,[4] Monty Mythen,[5] Michael P W Grocott[6,7]

For numbered affiliations see end of article.

**Correspondence to**
Dr Daniel S Martin;
daniel.martin@ucl.ac.uk

## ABSTRACT

**Introduction** Oxygen is the most commonly administered drug to mechanically ventilated critically ill adults, yet little is known about the optimum oxygen saturation ($SpO_2$) target for these patients; the current standard of care is an $SpO_2$ of 96% or above. Small pilot studies have demonstrated that permissive hypoxaemia (aiming for a lower $SpO_2$ than normal by using a lower fractional inspired oxygen concentration ($FIO_2$)) can be achieved in the critically ill and appears to be safe. This approach has not been evaluated in a National Health Service setting. It is possible that permissive hypoxaemia may be beneficial to critically ill patients thus it requires robust evaluation.

**Methods and analysis** Targeted OXygen therapY in Critical illness (TOXYC) is a feasibility randomised controlled trial (RCT) to evaluate whether recruiting patients to a study of permissive hypoxaemia is possible in the UK. It will also investigate biological mechanisms that may underlie the links between oxygenation and patient outcomes. Mechanically ventilated patients with respiratory failure will be recruited from critical care units at two sites and randomised (1:1 ratio) to an $SpO_2$ target of either 88%–92% or ≥96% while intubated with an endotracheal tube. Clinical teams can adjust $FIO_2$ and ventilator settings as they wish to achieve these targets. Clinical information will be collected before, during and after the intervention and blood samples taken to measure markers of systemic oxidative stress. The primary outcome of this study is feasibility, which will be assessed by recruitment rate, protocol adherence and withdrawal rates. Secondary outcomes will include a comparison of standard critical care outcome measures between the two intervention groups, and the measurement of biomarkers of systemic oxidative stress. The results will be used to calculate a sample size, likely number of sites and overall length of time required for a subsequent large multicentre RCT.

**Ethics and dissemination** This study was approved by the London - Harrow Research Ethics Committee on 2 November 2017 (REC Reference 17/LO/1334) and received HRA approval on 13 November 2017. Results from this study will be disseminated in peer-reviewed journals, at medical and scientific meetings, in the NIHR Journals Library and patient information websites.

**Trial registration number** NCT03287466; Pre-results.

### Strengths and limitations of this study

► This is the first study of permissive hypoxaemia in critically ill patients in a National Health Service setting.
► It is a small randomised controlled trial (RCT) to assess feasibility and not the efficacy of the intervention.
► The study will compare levels of biomarkers of systemic oxidative stress between the two intervention groups.
► It will provide valuable information to enable the design of future large-scale RCTs.

## INTRODUCTION

In the UK there are over 190 000 admissions to adult critical care units each year (www.icnarc.org). Approximately 40% of these patients will require mechanical ventilation and the mortality rate in this group is approximately 30%.[1 2] Hypoxaemia is common among this cohort of patients and we lack evidence-based guidelines for their management, particularly regarding what levels of arterial oxygenation are acceptable or optimal. It has been proposed that attempting to fully reverse hypoxaemia in critically ill patients may pose a greater risk of harm than allowing moderate hypoxaemia to persist, a concept called permissive hypoxaemia.[3] The premise behind permissive hypoxaemia is that the interventions used to correct hypoxaemia may themselves cause harm, in particular high concentrations of inspired oxygen, therefore safely minimising their use could be beneficial.[4–7]

Oxygen has the potential to cause harm when used in high concentration, primarily via its toxic effect on the lungs.[8–10] Reactive oxygen species (ROS) released mainly from the inner mitochondrial membrane

during oxidative phosphorylation serve an essential role in cellular signalling but in excess these highly reactive molecules are able to destroy lipids, proteins and DNA. Their rate of release is determined by cellular oxygen tension[11] and the extent of damage caused by them can be measured by evaluating biomarkers of tissue degradation.[12] The lung parenchyma is particularly susceptible to oxygen toxicity in critically ill patients as a result of being exposed to high concentration oxygen during mechanical ventilation. The threshold above which harm may be caused (in terms of concentration and duration of exposure) in critically ill patients is unclear, but since lung injury is common in this patient cohort the threshold may well be lower than in other patients or healthy volunteers. During critical illness the propagation of proinflammatory pathways, with the activation of leucocyte and vascular endothelial responses further increase the ROS burden.[13] This depletes endogenous antioxidants, which normally regulate ROS homoeostasis.[14] As a consequence, oxidative stress is a key mechanism of injury in systemic multiorgan failure, and has been linked to increase in morbidity and mortality in critical illness.[15 16]

Oxygen is a drug with a relatively narrow therapeutic index. It should therefore be prescribed, administered and monitored in a manner comparable to other drugs that have toxic side effects. There appears, however, to be wide variation in practice regarding its use and opinions about oxygenation in the critically ill.[17 18] This is perhaps the result of a paucity of evidence from robust clinical trials; a somewhat surprising situation given that almost every patient admitted to a critical care unit will receive supplementary oxygen. The traditional teaching that hypoxaemia must be avoided at all costs, may have led to a disregard to the potential harm caused by excessive oxygen, and this requires evaluation.

A small number of studies have begun to explore permissive hypoxaemia as a viable treatment strategy in the critically ill, primarily assessing feasibility and safety. In the first study of its kind, 105 mechanically ventilated patients were assessed in a before (n=51) and after (n=54) design.[19] Following a period of standard oxygenation practice in a single centre (aiming for normal or high blood oxygen levels), a practice change was initiated in which oxygen saturation ($SpO_2$) was maintained at 90%–92%. The authors of this study concluded that the conservative oxygen therapy intervention was feasible and free of adverse biochemical, physiological or clinical outcomes. A comparable strategy was used on a much larger scale in a two-stage model, moving from normal oxygenation to an $SpO_2$ of 92%–95%.[20] These authors reported that mechanical ventilation time was significantly lower during both study (lower $SpO_2$) phases compared with baseline. The adjusted intensive care unit (ICU) mortality and ICU-free days did not significantly differ between study phases but mortality decreased in reference to baseline for both of the low $SpO_2$ phases. In the first multicentre randomised controlled trial (RCT) of permissive hypoxaemia a total of 103 mechanically ventilated patients

were allocated to either a conservative oxygenation group ($SpO_2$ of 88%–92%) or a liberal oxygenation group ($SpO_2$ of greater than or equal to 96%).[21] The purpose of the study was to confirm feasibility and this was demonstrated, along with no excess of adverse events, in the low $SpO_2$ group. The most recently published trial of permissive hypoxaemia was a single-centre RCT that compared $SpO_2$ targets of 94%–98% versus 97%–100%.[22] The primary outcome of this study was ICU mortality, and the values reported were: 11.6% in the conservative group and 20.2% in the conventional group, giving an absolute risk reduction of 8.6% (1.7%–15.0%). This study had a number of limitations[23] but still adds weight to the argument that permissive hypoxaemia appears to not be harmful and may be of benefit.

A factor of great importance to the design of future studies is selecting the correct 'standard' treatment group, in order that the comparison to an intervention of lower oxygenation is valid and meaningful. Different studies have approached this in different ways (either by aiming for an oxygenation target or by determining the administered concentration of oxygen). We hope this feasibility will evaluate our selected methodology and allow us to compare it to other approaches that have been used. We also hope the results of this study will allow us to understand more about other issues specific to critically ill patients such as the concomitant presence of anaemia, low cardiac output and acute respiratory distress syndrome and chronic obstructive pulmonary disease.

## METHODS
The trial was designed according to the Standard Protocol Items: Recommendations for Interventional Trials statements.[24]

### Trial aim and objectives
*The Targeted OXYgen therapY in Critical illness* (TOXYC) study aims to determine whether reducing the $SpO_2$ target in patients requiring mechanical ventilation is feasible (in terms of participant recruitment and delivery of the intervention) in a National Health Service setting. In doing this we hope to inform future investigators who wish to construct larger trials in this field. The objectives are to construct an RCT of conventional oxygenation versus permissive hypoxaemia, identify any potential barriers to research in this field and explore biological mechanisms that may explain the proposed benefits from the intervention. The project was favourably supported at the UK Critical Care Research Forum in 2016.

### Primary outcome measure
The primary outcome of this study is feasibility, which will be assessed in the following ways: (1) The ability to recruit patients (recruitment rate). (2) Protocol deviations. (3) Rate of withdrawal from the study (in both the intervention and control groups). (4) The reasons for any withdrawal from the study.

Feasibility of recruitment will be evaluated by monitoring patient screening and subsequent agreement to participate, along with any withdrawal of consent during or after the study. Implementation of the study protocol will be evaluated by analysing adherence to oxygenation targets and completion of the treatment without protocol deviations. Reasons for withdrawal will be assessed by the trial management group at the end of the study to assess whether there are common themes that can be addressed in the future.

## Secondary outcome measures

Measurements of oxidative stress (including 4-hydroxynonenal, protein carbonyls, total antioxidant capacity and glutathione reductase) will be made in blood samples taken from participants, in order to understand the potential biological mechanisms that link blood oxygen levels to clinical outcomes. In addition, routine clinical data and outcome measures will be collected from the participants to assess any adverse effects caused by the intervention. Finally, length of critical care stay, length of hospital stay, and survival at critical care unit discharge, 30 days and 90 days will be collected. This information will be essential for the design of future larger trials.

## Trial design

TOXYC, a multicentre RCT, which will be conducted at two sites, is a trial of targeted oxygen therapy in adult critically ill patients receiving mechanical ventilation via an endotracheal tube. Sixty patients will be allocated on a 1:1 basis to either a normal $SpO_2$ target group or a lower than normal $SpO_2$ target group. A flow diagram of the study is shown in figure 1. Recruitment began in February 2018 and is planned to continue for 15 months.

## Selection of participants

### Screening

Screening will occur as part of routine research activity on the two critical care units involved in the study. Research nurses will use medical notes to determine initial suitability for the study, according to the inclusion and exclusion criteria. No additional tests or examinations will be required to ascertain whether patients are eligible for the study. Screening will occur as patients are admitted to the critical care units to minimise the time from admission to enrolment.

### Inclusion criteria

► Unplanned admission to a critical care unit.
► 18 years of age and above (no upper age limit).
► Respiratory failure forms part of the admission diagnosis.
► Enrolled within 24 hours of admission (if already intubated) or within 24 hours of intubation (if intubated on ICU).
► The patient is expected to receive mechanical ventilation for >72 hours.

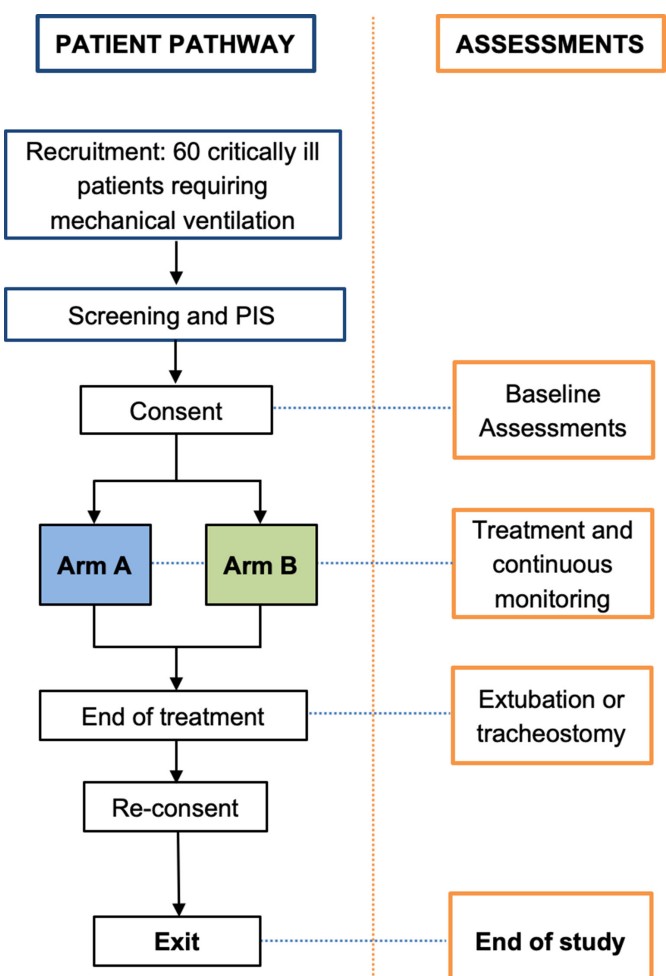

Figure 1 Study flow diagram. PIS, patient information sheet.

### Exclusion criteria

► Admission following surgery (elective or unplanned).
► Those patients expected to die within 24 hours of admission to ICU.
► Pregnant women.
► Admission postcardiac arrest.
► Patients with chronic lung disease known (or highly suspected) to have baseline $SpO_2$ in the range of the intervention arm (ie, 88%–92%).
► Admission post-trauma (including traumatic brain injury).
► Known sickle cell trait or disease.
► Ongoing significant haemorrhage or profound anaemia.
► Severe peripheral vascular disease.
► Severe pulmonary hypertension.
► Other medical conditions where mild hypoxaemia would be contraindicated.
► Patients participating in other interventional clinical trials.

## Enrolment

### Consent

Due to the severity of illness of the patients being recruited to this study, and the use of sedative drugs that

are required for mechanical ventilation, it is unlikely that potential participants will have capacity. Should a potential participant be deemed to have capacity they will be approached by the research team, given a patient information sheet (PIS) and then provided with an opportunity to ask questions. After an appropriate length of time, the research team will seek informed consent from the patient if they wish to participate.

If the patient lacks capacity to provide informed consent, a Personal Consultee (PeC) will be appointed to represent them. This could be the patient's next of kin, a relative or close friend with whom to discuss the patient's participation in the trial. The research team will seek the PeC's opinion as to whether the patient would wish to take part in the trial, providing for them an appropriate version of the ethically approved PIS. If the PeC believes the patient would have wanted to participate in the study (or would not have objected to it), they will be asked to sign a PeC agreement form. If there is no PeC present or immediately available in person, opinion may be sought from a suitable person via the telephone and then a telephone agreement form completed by a member of the research team.

If there is no identifiable PeC for a potential participant then they will be provided with a Professional Consultee (PrC), who is completely independent of the study. Their opinion will be sought as to whether it is appropriate for the patient to be enrolled into the study. Opinion will be sought in the same manner as for the PeC, involving the appropriate version of the ethically approved PIS.

Adequate time will be given for consideration by the patient or PeC/PrC to consider the information in the PIS and ask questions. The research team will record when the PIS has been given to the patient or their consultee. Due to the nature of this patient cohort (critically ill patients requiring substantial organ support due to the severity of their illness) the length of time from identifying a potential participant to initiating the intervention is likely to be less than 24 hours. This is to avoid dilution of the intervention or control effect prior to its commencement.

If a participant who lacked capacity at the point of recruitment subsequently becomes able to provide informed consent (because they gain capacity on recovery from their illness), they will be informed about their participation in the study, provided with a PIS and asked whether they would be willing to provide retrospective consent. At this point the participant will be given the opportunity to withdraw from the study and to decide if the data (and blood samples) collected from them can be included in the final analysis.

All patient consent and consultee agreement procedures will adhere to the Mental Capacity Act (2005).

### Randomisation
Randomisation will be carried out online after a patient has been recruited to the study and an agreement form or consent form has been signed. It will be conducted in a 1:1 manner for the intervention and control groups, stratified by study site, using random permuted blocks of different sizes. The process of randomisation will be conducted online (www.sealedenvelope.com).

### Withdrawal
Participants will be withdrawn from the study if:
► The responsible clinician deems it inappropriate for treatment to continue due to a change in the patient's condition.
► The chief investigator or delegated member of the research team deems it inappropriate for treatment to continue due to a change in the patient's condition.
► Agreement for the patient to participate in the study is withdrawn by the PeC or PrC.
► The patient regains capacity and chooses to withdraw from the study.

As this is a feasibility study, patients withdrawn from the study will not be replaced, but the reason for withdrawal will be recorded.

### Trial treatment
#### Intervention
The intervention is a more conservative use of oxygen via the ventilator to achieve an $SpO_2$ of 88%–92%, lower than normal practice in most critical care units in the UK. The intervention will be delivered by the participant's clinical team, which will consist of the critical care doctors and nurses at the two study centres. These teams will be provided with guidance to help keep participants within their target $SpO_2$ (supplementary material 1) but this will not be protocolised. Due to the nature of the intervention, neither the research nor the clinical teams can be blinded to participant group allocation.

#### Comparator
The control group will receive oxygen to maintain an $SpO_2$ at 96% or above (standard care). As per the intervention group, guidance will be provided to the clinical team to help maintain participants within their target $SpO_2$ (online supplementary material 2).

#### Duration of treatment
The aim is for the intervention to be commenced as soon as possible after admission to the critical care unit (following enrolment) and end following removal of the participant's endotracheal tube. Specific treatment end points for both groups would therefore include (1) Extubation. (2) Formation of a tracheostomy. (3) Transfer to another critical care unit. (4) Death. The research team will review enrolled participants daily to monitor adherence to $SpO_2$ targets and provide bedside advice where required. No targets or limitations will be set for arterial $PaO_2$ of oxygen ($PaO_2$) or carbon dioxide ($PaCO_2$). Should a patient in the study be transferred out of the ICU for a short period of time (eg, for an investigation or intervention) the protocol will be paused until their return. While out of the ICU the clinical team will be in control of the patient's oxygenation. Should a patient's

condition deteriorate to such an extent that either the clinical or research team feel it not in the patient's best interest to continue in the study, they will be withdrawn from it at that point.

### Standard clinical management

Aside from the designated $SpO_2$ targets, all other aspects of care will remain the same between the intervention and control groups. Regular arterial blood gases should be taken during the trial period, according to local clinical guidelines; no additional arterial blood gases will be necessary for the purpose of the study.

## Data collection

Data will be collected from various sources including the participant's medical records, bedside charts and hospital computer systems. Data will be collected from these primary sources by members of the research team and entered into an electronic clinical record form (eCRF).

### Baseline data collection

► Patient demographics: age, gender, height and weight.
► Cause of respiratory failure (diagnosis).
► The presence of any chronic diseases.
► Acute Physiology and Chronic Health Evaluation (APACHE) II Score (and its components).
► Sequential Organ Failure Assessment (SOFA) Score (and its components).
► *Respiratory measurements*: $PaO_2$, $PaCO_2$, pH, $SpO_2$, $FIO_2$, ventilator settings and measures.
► *Cardiovascular measurements*: blood pressure, heart rate, cardiac rhythm, vasopressor/inotrope dose, fluid balance.
► *Renal measurements*: creatine, urine output in the past 24 hours, the need for renal replacement therapy.
► *Hepatic measurements*: transaminases, blood clotting values and bilirubin.
► Blood lactate concentration.

### Subsequent data collection during treatment

► Most measures will be taken daily, except for those specifically related to oxygenation, which will be collected hourly, to permit detailed analysis of compliance to blood oxygenation target.
► Time to extubation or detachment from mechanical ventilation, and mechanical ventilation-free days on ICU.
► Adverse events occurring during the study period.

### Follow-up

► Length of ICU stay.
► Length of hospital stay.
► 30-day and 90-day survival rates, and days alive out of hospital.
► Adverse events.

## Data management

This trial will use an eCRF and trial data will be entered into an approved, protected database (https://www.elsevier.

com/solutions/macro). Access to the eCRF system will only be provided to staff with relevant authority. Participants will be given a unique subject number and subject identifier. Data will be entered under this identification number onto the central database stored on the servers. The database will be password protected and only accessible to members of the TOXYC study team and external regulators if requested. At site, access will only be granted to staff with permission on the delegation log, and after training. The servers are protected by firewalls and are patched and maintained according to best practice. The physical location of the servers is protected by closed-circuit TV and security door access. The database software provides a number of features to help maintain data quality, including; maintaining an audit trail, allowing custom validations on all data, allowing users to raise data query requests and search facilities to identify validation failure/missing data.

The identification, screening and enrolment logs, linking participant identifiable data to the pseudoanonymised subject numbers will either be held in written form in a locked filing cabinet or electronically in password protected form on hospital computers. After completion of the study the identification, screening and enrolment logs will be stored securely by the sites for 20 years.

## Sample collection, storage and processing

Blood samples will be taken from participants in order to evaluate oxidative stress. Samples will be taken from an indwelling arterial catheter that is already present in the patient as part of routine critical care. Blood will be processed at each of the two centres according to a defined standard operating procedure, and then stored at −80°C. These blood samples will be taken at baseline (shortly after the patient has been recruited into the study but prior to their treatment being commenced), and on days 2, 3, 5 and 10 after recruitment. A number of biomarkers of oxidative stress will be measured, including: 4-hydroxyl-2-nonenal, protein carbonyls, total antioxidant capacity and glutathione reductase.

## Safety monitoring

All adverse events will be recorded in the medical records and reported to the appropriate body; either online to SITU or by email to the sponsor. Table 1 shows a basic list of expected adverse events that will be recorded within the patient's eCRF and medical notes, but the sponsor will not be informed. Adverse events and serious adverse events will be summarised descriptively by the interventional group at the end of the study.

## Trial monitoring and oversight

The TOXYC Trial will report to a data monitoring committee, and a trial steering committee will be appointed to provide study oversight on behalf of the sponsor and funder. Day-to-day management of the trial will be the responsibility of the trial coordinator with oversight from the trial management group. Permission

**Table 1** A list of expected adverse events that may occur during the course of the study

| Respiratory | Cardiovascular | Haematological |
|---|---|---|
| ► Reintubation<br>► Arterial desaturation<br>► Pneumothorax<br>► Pleural effusion<br>► Pneumonia<br>► Pulmonary embolism | ► Arrhythmia<br>► Hypotension<br>► Requirement for inotropic support | ► Anaemia<br>► Low platelet count<br>► High leucocyte count |
| **Renal** | **Gastrointestinal** | **Neurological** |
| ► Acute kidney injury<br>► Requirement for renal support<br>► Hyperkalaemia | ► Diarrhoea<br>► Vomiting<br>► Failure to absorb enteral feed | ► Delirium/agitation |

for protocol amendments will be sought via the sponsor and, if deemed necessary, the research ethics committee.

### Statistical design

No sample size calculation was performed to determine the number of participants required for this trial. The reason for this was that it is a feasibility study in which no formal comparative analyses are planned. The primary and secondary outcome measures will be presented using summary statistics (eg, means, SD, medians, proportions). Missing data, non-compliers and withdrawals will be looked at in detail to determine if there is any evidence of bias.

Those analysing the data will be blinded to specific group allocation. A CONSORT diagram will be completed, summarising the number of patients eligible for the study, the number randomised in each arm, and enumerating in detail those not approached (with reasons), the number of withdrawals (with reasons) overall and per arm. Recruitment rate (overall, per site and peak recruitment rate per month) will be determined. Monthly and cumulative accrual graphs will be constructed. Baseline characteristics of randomised participants will be summarised, including gender, age, height, weight, details of medical history, preintervention APACHE II and SOFA scores. Mean and SD, or median and IQR, will be calculated as appropriate. For the secondary objectives, summary statistics on respiratory, cardiovascular, renal and hepatic measurements will be calculated at the appropriate time points (hourly or daily). Length of stay in the critical care unit and in hospital will be summarised. Thirty-day and 90-day mortality rates will be calculated, and 'days alive and out-of-hospital' determined for each patient and summarised using appropriate measures. Compliance will be assessed. For each patient, the proportion of time spent within the randomisation-determined $SpO_2$ limits will be calculated, and summarised by treatment arm. Adverse events will be tabulated and grouped according to seriousness, severity and causality.

All variables will be checked for completeness and checked for the presence of outliers. Graphical depictions of results will be prepared, both on a per-patient basis (especially for the longitudinal data such as hourly oxygen measurement) and grouped by intervention. Frequency distribution curves will be shown where appropriate (especially for the 'length of stay' measurements).

No hypothesis testing is envisaged for this feasibility study. The results will be used to calculate a sample size, likely number of sites and overall length of time required for a subsequent large multicentre RCT.

### Patient and public involvement

In 2014, the UK Intensive Care Society published the results of their James Lind Alliance Priority Setting Partnership.[25] The aim was to identify and prioritise unanswered questions about adult critical care that are important to people who have been critically ill, their families and the health professionals who care for them. One of the identified priorities for research was: *'What is the best way of preventing damage to the lungs of patients receiving respiratory support (ventilation)?'*. This study addresses this publicly driven need, by assessing a treatment strategy that has the potential to reduce iatrogenic lung injury to patients on a ventilator and therefore improve survival. At the Royal Free Hospital critical care unit we have a growing group of patients and relatives that is willing to assist with the development of research. This group was formed and is managed by our research team at the Royal Free Hospital. A volunteer member of the public from this group agreed to assist us throughout the study, from the application for funding to dissemination of the findings. This person was a coapplicant on the grant application and is an invited member of the Trial Steering Committee. We hope that including a member of the public in the design and delivery of the study will improve the experience of participants in this an future clinical research projects.

### CONCLUSION

The results of this feasibility trial will inform researchers about the ability to conduct a study evaluating permissive hypoxaemia in critically ill patients. It will provide information about recruitment rates in UK critical care units and help to identify any barriers to future research. Furthermore, results from

oxidative stress marker analysis may highlight biological markers of importance in the pathway between oxygen administration and patient outcomes.

**Author affiliations**
[1]Critical Care Unit, Royal Free Hospital, London, UK
[2]Division of Surgery and Interventional Science, Royal Free Hospital, University College London, London, UK
[3]Surgical and Interventional Trials Unit, Division of Surgery and Interventional Science, University College London, London, UK
[4]Manchester Academic Health Sciences Centre, Salford Royal Foundation NHS Trust, Salford, UK
[5]Anaesthesia and Critical Care, University College London Hospitals National Institute of Health Research Biomedical Research Centre, London, UK
[6]Integrative Physiology and Critical Illness Group, Clinical and Experimental Sciences, Faculty of Medicine, University of Southampton, Southampton, UK
[7]Critical Care Research Group, Southampton NIHR Biomedical Research Centre, University Hospital Southampton, Southampton, UK

**Contributors** DSM: Conceived idea, designed study, obtained funding for RCT, obtained funding for mechanistic component, obtained ethics approval, will be CI for study and PI at Royal Free, wrote manuscript, revised manuscript. CB-G: obtained ethics approval, responsible for trial governance, wrote manuscript, revised manuscript. NM: obtained ethics approval, responsible for trial governance, revised manuscript. GJ: obtained funding for mechanistic component, lead for biochemical analysis, revised manuscript. IP: responsible for data collection, revised manuscript. JS: involved in sample processing, revised manuscript. NW: obtained ethics approval, responsible for trial governance, responsible for data analysis and statistics, wrote manuscript, revised manuscript. MMcN: lead research nurse for study, responsible for screening patients and collecting data, revised manuscript. RO'D: obtained funding for RCT, involved in study design, revised manuscript. MM: Conceived idea, designed study, obtained funding for RCT, revised manuscript. MPWG: Conceived idea, designed study, obtained funding for RCT, PI at Southampton General Hospital, wrote manuscript, revised manuscript.

**Funding** This work was supported by the National Institute for Health Research (PB-PG-0815-20006) and Royal Free Charity.

**Competing interests** DSM, MM and MPWG are directors of a company developing an oxygen delivery device (Oxygen Control Ltd). DSM has received honoraria and consultancy fees from Siemens Healthcare, Masimo, Deltex and Edwards Lifesciences. MPWG is the National Specialty Lead for Anaesthesia, Perioperative Medicine and Pain within the UK National Institute of Heath Research Clinical Research Network, an elected council member of the Royal College of Anaesthetists and serves on the board of the Evidence Based Perioperative Medicine (EBPOM) social enterprise and the medical advisory board of Sphere Medical Ltd. MPWG has received honoraria for speaking and/or travel expenses from Edwards Lifesciences, Fresenius-Kabi, BOC Medical (Linde Group), Ely-Lilly Critical Care, and Cortex GmBH. MPWG is executive chair of the Xtreme-Everest Oxygen Research Consortium and joint Editor-in-Chief of the journal Perioperative Medicine. MM is a consultant for Baxter, Edwards Lifesciences and Deltex; his University Chair is supported by Smiths Medical; Elected Council Member Royal College of Anaesthetsists; Editorial Board BJA and Critical Care; Founding Editor-in-Chief Perioperative Medicine.

**Patient consent for publication** Not required.

**Ethics approval** London - Harrow Research Ethics Committee.

**Provenance and peer review** Not commissioned; externally peer reviewed.

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
