## [Reviewer comments · BMJ Open]

ARTICLE DETAILS

TITLE (PROVISIONAL)	Protocol for a feasibility randomised controlled trial of targeted oxygen therapy in mechanically ventilated critically ill patients
AUTHORS	Martin, Daniel S.; Brew-Graves, Chris; McCartan, Neil; Jell, Gavin; Potyka, Ingrid; Stevens, Jia; Williams, Norman; McNeil, Margaret; O'Driscoll, Ronan; Mythen, Monty; Grocott, Michael P. W.

VERSION 1 – REVIEW

REVIEWER	Huaiwu, He PUMCH, CHINA
REVIEW RETURNED	05-Feb-2018

GENERAL COMMENTS	1. A low SpO₂(92%) has been suggested for the ARDS patients. So, the ARDS patients might be confounded in this study. 2. The most risk of low SpO₂ was tissue hypoxia. If the patients had a shock condition, a too low SpO₂ would be improper. The related content should be added to the standards of patients recruitment. 3. The titration of SpO₂ should be individualized in the future. It might be important to identify who could benefit from low SpO₂. So, more detail of the patient's potential response should be included during the titration of SpO₂. 4. The calculation of sample should be revealed.
---

REVIEWER	Satoshi Suzuki Department of Anesthesiology and Resuscitology, Okayama University Hospital, Okayama, Japan
REVIEW RETURNED	21-Mar-2018

GENERAL COMMENTS	This manuscript outlines the protocol for the randomised controlled trial to test the feasibility and safety of conservative oxygen therapy in mechanically ventilated critically ill patients at 2 centres in the UK. I think the topic is interesting and important to the UK critical care setting. I have only a few comments. 1. The authors should describe the date when the study will start/started. 2. P18L7: I think the authors should perform comparative analysis to assess the safety of the intervention. The authors should describe how to determine if the intervention is safe.
--

REVIEWER	Rakshit Panwar John Hunter Hospital, Australia
REVIEW RETURNED	26-Mar-2018

GENERAL COMMENTS	Thank you for this opportunity to review your work. This paper describes the methodology for a pilot RCT that aims to assess
--

feasibility of targeted oxygen therapy in mechanically ventilated critically ill patients. This is no doubt an extremely important step to pave way for a large definitive RCT, which will be an invaluable contribution to the literature in this area. However, there are several major concerns about the protocol. Please see my comments below:

Major:

1. I wonder if eligibility criteria are too restrictive, thereby limiting the potential patient base. For example,
 - a. Some patients might develop respiratory failure after admission to ICU, as progression of an initial lung insult is not uncommon 1. In a large observational study, about one-third of patients developed acute lung injury or ARDS at a median of 3 days after the ICU admission 2. Possibly a subclinical preexisting insult to the lung may alter susceptibility to oxygen-induced lung damage and predispose to developing ARDS in response to hyperoxia 3-7. It could be useful to enroll such patients too.
 - b. It is often difficult to gauge, even for an astute clinician, for how long a patient might need invasive mechanical ventilation. 72 hours of invasive mechanical ventilation does seem to be a relatively long period to be able to estimate prospectively and reasonably accurately. There may be some leeway to relax this threshold requirement, otherwise a fair number of potentially eligible patients might be missed.
 - c. Hyperoxemia has been shown to be independently associated with an increased in-hospital mortality among patients who are admitted after cardiac arrest 8,9. It is probably worth including these patients too.
 - d. Likewise, hyperoxemia is also shown to be independently associated with higher in-hospital mortality among ventilated patients admitted with traumatic brain injury 10. The brain trauma foundation guideline recommends avoiding hypoxia (PaO₂ <60 mmHg or SaO₂ <90%) among patients with neurotrauma 11. These thresholds are well within the intervention range of this pilot RCT.

Would it be worth including all such patients and pre-specify these subgroups analyses in the larger definitive trial?

1. I am intrigued to learn that a SpO₂ target range of 98-100% can be considered as 'standard care' at NHS hospitals. I think over last few years the standard care in relation to oxygenation has certainly shifted towards avoiding use of too much oxygen. The concern with using a default SpO₂ target of 96-100% as 'control' is that it may end up being somewhat dissimilar to the actual standard care by the time the trial is completed. For example, not many contemporary intensivists would be comfortable aiming for SpO₂ >96% while FiO₂ is >0.60 or >0.70 in a ventilated patient with respiratory failure, particularly when all the ARDS RCTs targeted a SpO₂ range of 88-95% 12-14. Ventilating at a FiO₂ of 1 is demonstrably harmful as recently shown in a large multicenter RCT 15. Those surveys that have been referenced seem outdated in context of several recent publications highlighting the potential harms of hyperoxia 15-18. This is also reflected in the 2017 British Thoracic Society recommendations to aim for a SpO₂ target range of 94-98% among critically ill patients 19; or in other words, to avoid SpO₂ >98%, the usual threshold for hyperoxia. I understand it is important to achieve adequate separation in the overall cohort, but this perhaps needs to be balanced against the risk of having a somewhat "artificial" standard care arm. It might be worth considering either having a true standard care arm, in which

	clinicians are free to prescribe oxygen as they would normally do in their day-to-day practice, or limit the FiO2 titration range for the purpose of this study to 0.21-0.80 as done in a previous pilot RCT 20. 2. It would be ideal to further minimize the pre-randomization duration of mechanical ventilation. It appears that the intention is to keep it under 24 hours, but perhaps keeping it under 12 hours would go a fair way in ascertaining that patients assigned to the 88-92% SpO2 target would not be exposed to standard liberal oxygenation for prolonged periods prior to randomization. This would minimize "contamination". 3. The rationale to limit exposure to high SpO2, that might occur in the standard practice, is also applicable to the period after extubation. Therefore, it may be questioned why the study intervention should cease following the end of invasive mechanical ventilation. This could run a risk of exposing patients assigned to 88-92% SpO2 arm to liberal oxygen targets after extubation, potentially diluting the 'intervention effect', if there is any. The use of high flow oxygen through nasal cannula is increasing in ICUs. In my view, the study intervention should perhaps continue until the patient is liberated from all respiratory support, rather than just mechanical ventilation. 4. The accuracy of pulse oximetry depends on the perfusion adequacy or the pulse amplitude of the selected monitoring site. Since the FiO2 in this trial will be titrated to the assigned SpO2 range measured by pulse oximetry, the investigators do not describe whether, or how, perfusion adequacy or the pulse amplitude will be checked amongst participants. Would health care professionals, who are caring for participants, try different monitoring sites to check where the best pulse amplitude is obtained, and/or check SpO2 against CO-oximetry measured SaO2 to ensure the accuracy of the targeted parameter? For example,  a. Use of restraints in ICU is not that uncommon. It is an established factor that may result in an inherent inaccuracy if pulse oximetry is performed in a restrained limb. b. Different fingers may have different perfusion indices. c. An indwelling line in the supplying artery may potentially interfere with the perfusion index. Will the pulse oximeter probes exclusively placed on the limb without an invasive arterial cannula? 5. In addition, pulse oximeters can be of various types. Some use transmission method, some use reflectance, some are clip-on types, whereas others are wrap-arounds or wear-on rubber types. These may all have different biases and agreements limits when compared to a true SaO2. Will the investigators mandate the use of single type pulse oximeters for all participants? Or will this data be also collected and presented for the participants? 6. It is not that clear from the manuscript what specific tests for oxidative stress will be performed. Are you considering any global parameter for oxidative stress such as the plasma redox potential based on CyS-CySS ratio, derived from the Nernst equation ($Eh_{CyS/CySS} = -250 + 30 \log ([CyS]/[CySS])$) 21-23. Minor:
--	---

7. It is well known that the effects of O₂ on the lungs and brain are considered as part of a patho-physiological continuum in which a common factor is Nitric oxide (NO) and its interaction with other reactive species 24 that can produce a toxic oxidant called peroxynitrite, which may result in a selective nitration of tyrosine in proteins to create nitrotyrosine 25. Since the occurrence of protein tyrosine nitration under disease conditions is an established marker of NO-dependent oxidative stress 26, will the investigators consider measuring the plasma levels of protein bound 3-nitrotyrosine levels in this study?

8. Will the assigned SpO₂ targets remain in place at times of hemodynamic instability or during trips outside ICU for therapeutic or diagnostic purposes?

9. It seems unclear if the primary and secondary outcome measures will be presented for each arm separately. If this is not planned, that is understandable for a feasibility study. However, it would be important to show whether there was a meaningful separation in the achieved SpO₂, SaO₂, PaO₂, and FiO₂. Is this part of the planned analysis?

10. If the comparative analysis is not planned, then will the patients enrolled in this pilot-RCT be included in the larger RCT later on?

11. Perhaps the study posters for both arms should clearly emphasize that "the lowest possible FiO₂ should be used to achieve the target SpO₂ range". This would be helpful in achieving best possible separation.

References

1. Esteban A, Anzueto A, Frutos F, et al. Characteristics and outcomes in adult patients receiving mechanical ventilation: a 28-day international study. *Jama*. 2002;287(3):345-355.
2. Brun-Buisson C, Minelli C, Bertolini G, et al. Epidemiology and outcome of acute lung injury in European intensive care units. Results from the ALIVE study. *Intensive Care Med*. 2004;30(1):51-61.
3. Aggarwal NR, D'Alessio FR, Tsushima K, et al. Moderate oxygen augments lipopolysaccharide-induced lung injury in mice. *American journal of physiology. Lung cellular and molecular physiology*. 2010;298(3):L371-381.
4. Altemeier WA, Sinclair SE. Hyperoxia in the intensive care unit: why more is not always better. *Curr Opin Crit Care*. 2007;13(1):73-78.
5. Knight PR, Kurek C, Davidson BA, et al. Acid aspiration increases sensitivity to increased ambient oxygen concentrations. *Am J Physiol Lung Cell Mol Physiol*. 2000;278(6):L1240-1247.
6. Matthay MA, Zimmerman GA. Acute lung injury and the acute respiratory distress syndrome: four decades of inquiry into pathogenesis and rational management. *Am J Respir Cell Mol Biol*. 2005;33(4):319-327.
7. Thiel M, Chouker A, Ohta A, et al. Oxygenation inhibits the physiological tissue-protecting mechanism and thereby exacerbates acute inflammatory lung injury. *PLoS Biol*. 2005;3(6):e174.

8. Kilgannon JH, Jones AE, Shapiro NI, et al. Association between arterial hyperoxia following resuscitation from cardiac arrest and in-hospital mortality. *Jama*. 2010;303(21):2165-2171.
9. Kilgannon JH, Jones AE, Parrillo JE, et al. Relationship between supranormal oxygen tension and outcome after resuscitation from cardiac arrest. *Circulation*. 2011;123(23):2717-2722.
10. Rincon F, Kang J, Vibbert M, Urtecho J, Athar MK, Jallo J. Significance of arterial hyperoxia and relationship with case fatality in traumatic brain injury: a multicentre cohort study. *Journal of neurology, neurosurgery, and psychiatry*. 2014;85(7):799-805.
11. The Brain Trauma Foundation. The American Association of Neurological Surgeons. The Joint Section on Neurotrauma and Critical Care. Resuscitation of blood pressure and oxygenation. *J Neurotrauma*. 2000;17(6-7):471-478.
12. Brower RG, Lanken PN, MacIntyre N, et al. Higher versus lower positive end-expiratory pressures in patients with the acute respiratory distress syndrome. *N Engl J Med*. 2004;351(4):327-336.
13. Wiedemann HP, Wheeler AP, Bernard GR, et al. Comparison of two fluid-management strategies in acute lung injury. *N Engl J Med*. 2006;354(24):2564-2575.
14. Acute Respiratory Distress Syndrome N, Brower RG, Matthay MA, et al. Ventilation with lower tidal volumes as compared with traditional tidal volumes for acute lung injury and the acute respiratory distress syndrome. *N Engl J Med*. 2000;342(18):1301-1308.
15. Asfar P, Schortgen F, Boisrame-Helms J, et al. Hyperoxia and hypertonic saline in patients with septic shock (HYPER2S): a two-by-two factorial, multicentre, randomised, clinical trial. *Lancet Respir Med*. 2017;5(3):180-190.
16. Stub D, Smith K, Bernard S, et al. Air Versus Oxygen in ST-Segment-Elevation Myocardial Infarction. *Circulation*. 2015;131(24):2143-2150.
17. Damiani E, Adrario E, Girardis M, et al. Arterial hyperoxia and mortality in critically ill patients: a systematic review and meta-analysis. *Crit Care*. 2014;18(6):711.
18. Helmerhorst HJ, Roos-Blom MJ, van Westerloo DJ, de Jonge E. Association Between Arterial Hyperoxia and Outcome in Subsets of Critical Illness: A Systematic Review, Metaanalysis, and Meta-Regression of Cohort Studies. *Crit Care Med*. 2015.
19. O'Driscoll BR, Howard LS, Earis J, Mak V. British Thoracic Society Guideline for oxygen use in adults in healthcare and emergency settings. *BMJ Open Respir Res*. 2017;4(1):e000170.
20. Panwar R, Hardie M, Bellomo R, et al. Conservative versus Liberal Oxygenation Targets for Mechanically Ventilated Patients. A Pilot Multicenter Randomized Controlled Trial. *Am J Respir Crit Care Med*. 2016;193(1):43-51.
21. Jones DP, Go YM, Anderson CL, Ziegler TR, Kinkade JM, Jr., Kirlin WG. Cysteine/cystine couple is a newly recognized node in the circuitry for biologic redox signaling and control. *FASEB J*. 2004;18(11):1246-1248.
22. Jones DP, Carlson JL, Mody VC, Cai J, Lynn MJ, Sternberg P. Redox state of glutathione in human plasma. *Free radical biology & medicine*. 2000;28(4):625-635.
23. Netto LE, de Oliveira MA, Monteiro G, et al. Reactive cysteine in proteins: protein folding, antioxidant defense, redox signaling and more. *Comp Biochem Physiol C Toxicol Pharmacol*. 2007;146(1-2):180-193.

	24. Allen BW, Demchenko IT, Piantadosi CA. Two faces of nitric oxide: implications for cellular mechanisms of oxygen toxicity. Journal of applied physiology. 2009;106(2):662-667. 25. Beckman JS, Koppenol WH. Nitric oxide, superoxide, and peroxynitrite: the good, the bad, and ugly. The American journal of physiology. 1996;271(5 Pt 1):C1424-1437. 26. Radi R. Nitric oxide, oxidants, and protein tyrosine nitration. Proc Natl Acad Sci U S A. 2004;101(12):4003-4008.
--	--

VERSION 1 – AUTHOR RESPONSE

TOXYC RESPONSES

Reviewer: 1

Reviewer Name: Huaiwu, He

Institution and Country: PUMCH, CHINA

Please state any competing interests: None declared

Please leave your comments for the authors below

1. A low SpO₂(92%) has been suggested for the ARDS patients. So, the ARDS patients might be confounded in this study.

We would like to thank the reviewer for taking the time to read and comment on our manuscript. We have provided responses to the comments below and made changes to the original document.

Many thanks for highlighting this important issue. Whilst a lower SpO₂ has been suggested as advantageous for patients by some authors with ARDS this is not common practice nor part of any definitive guideline in the UK. The recent ATS/ESICM guidelines for the treatment of patients with ARDS made no reference to optimum oxygenation (1) so we presume there is equipoise on this topic. Thus, we feel a study to determine the correct oxygen targets in patients with respiratory failure is justified and those patients diagnosed with ARDS (a small proportion of the overall cohort receiving mechanical ventilation in ICU) will not confound our results. If clinicians in the recruiting centres feel unhappy about allowing a patient with ARDS to be entered into the study we will not enrol them and record this on the screening log.

We have added text in the introduction to signify that we have considered this issue.

1) Fan E et al. An Official American Thoracic Society/European Society of Intensive Care Medicine/Society of Critical Care Medicine Clinical Practice Guideline: Mechanical Ventilation in Adult Patients with Acute Respiratory Distress Syndrome. *Am J Respir Crit Care Med*. 2017 May 1;195(9):1253-1263.

2. The most risk of low SpO₂ was tissue hypoxia. If the patients had a shock condition, a too low

SpO₂ would be improper. The related content should be added to the standards of patients recruitment.

We agree that the combination of hypoxaemia and hypoperfusion may risk inadequate convective oxygen delivery to organs and tissues. The difficulty with this is that it remains unclear what threshold of convective oxygen delivery is critical to an individual patient, and individual organs. We also appreciate that excessive anaemia will also impact on the issue of oxygen delivery. It is for this reason that we have excluded patients in whom there is "ongoing significant haemorrhage or profound anaemia." We purposely did not set a numerical threshold on the level of anaemia but left this to the judgement of the research and clinical teams. If patients have a degree of shock to an extent that their life is threatened by low cardiac output, the clinical and research team can withdraw a patient from the study.

We have added text in the introduction to signify that we have considered this issue. And we have added text to the methods highlighting this.

3. The titration of SpO₂ should be individualized in the future. It might be important to identify who could benefit from low SpO₂. So, more detail of the patient's potential response should be included during the titration of SpO₂.

We agree that in a future study it would be very interesting to look in greater detail at individual responses to oxygen titration in order to begin to stratify oxygen therapy. However, our aim in this study was to assess feasibility and demonstrate safety. Unfortunately it is outside the remit of this current study to carry out any detailed analysis of individual oxygen responsiveness.

4. The calculation of sample should be revealed.

This is a feasibility study and the primary outcome measure will therefore be feasibility. As such no formal sample size calculation has been undertaken (2). The information gained from this study will be essential for adequately sizing future studies by providing standard deviations for the selected outcome variables. It has been suggested that a total study size of at least 30 is required to achieve success in a pilot or feasibility study.

2) Lancaster GA, Dodd S, Williamson PR. Design and analysis of pilot studies: recommendations for good practice. *J Eval Clin Pract* 2004; 10:307-312.

Reviewer: 2

Reviewer Name: Satoshi Suzuki

Institution and Country: Department of Anesthesiology and Resuscitology, Okayama University Hospital, Okayama, Japan

Please state any competing interests: None declared.

Please leave your comments for the authors below

This manuscript outlines the protocol for the randomised controlled trial to test the feasibility and safety of conservative oxygen therapy in mechanically ventilated critically ill patients at 2 centres in the UK. I think the topic is interesting and important to the UK critical care setting. I have only a few comments.

1. The authors should describe the date when the study will start/started.

Thank you for taking the time to read our manuscript and provide feedback. We have provided responses to the comments raised below and updated the manuscript accordingly.

We shall include the start and projected completion time for the study.

2. P18L7: I think the authors should perform comparative analysis to assess the safety of the intervention. The authors should describe how to determine if the intervention is safe.

This is a very valid point, however, in a study of this size we will be unable to conduct a formal statistical comparison of adverse events between the groups as the study is not powered to detect a difference. We shall report the number of adverse events in each group.

We are comparing what we believe to be standard UK practice with a degree of permissive hypoxaemia that has been identified as safe in the pilot studies that are cited in our introduction section. Therefore, we do not believe that we are exposing patients to any risks undue risk as a result of this study.

A full safety assessment can only be achieved in a larger multicentre study.

Reviewer: 3

Reviewer Name: Rakshit Panwar

Institution and Country: John Hunter Hospital, NSW,
Australia

Please state any competing interests: None declared

Please leave your comments for the authors below

Thank you for this opportunity to review your work. This paper describes the methodology for a pilot RCT that aims to assess feasibility of targeted oxygen therapy in mechanically ventilated critically ill patients. This is no doubt an extremely important step to pave way for a large definitive RCT, which will be an invaluable contribution to the literature in this area.

Thank you for providing such a comprehensive review of our manuscript. We very much value your comments and have responded to them below, with appropriate changes made to the manuscript.

However, there are several major concerns about the protocol. Please see my comments below:

Major:

1. I wonder if eligibility criteria are too restrictive, thereby limiting the potential patient base. For example,
 - a. Some patients might develop respiratory failure after admission to ICU, as progression of an initial lung insult is not uncommon ¹. In a large observational study, about one-third of patients developed acute lung injury or ARDS at a median of 3 days after the ICU admission ². Possibly a subclinical preexisting insult to the lung may alter susceptibility to oxygen-induced lung damage and predispose to developing ARDS in response to hyperoxia ³⁻⁷. It could be useful to enroll such patients too.

Thank you for raising this point. Our study was specifically designed to assess the intervention in intubated patients. Whilst we would also like to understand the feasibility and safety of the intervention in those patients not requiring intubation on ICU, it was outside the remit of this project. Shortly after opening our study to recruitment, we realised the very point you have highlighted though, i.e. that patients may be admitted to ICU and then deteriorate. We therefore immediately requested (and were granted) an amendment to the original protocol such that the inclusion criteria regarding this area now states: "Enrolled within 24 hours of admission (if already intubated) or within 24 hours of intubation (if intubated on ICU)." We hope this will widen the scope of our recruitment to include not only patients who arrive at the ICU intubated, but those who deteriorate and then require intubation some time after admission.

- b. It is often difficult to gauge, even for an astute clinician, for how long a patient might need invasive mechanical ventilation. 72 hours of invasive mechanical ventilation does seem to be a relatively long period to be able to estimate prospectively and reasonably accurately. There may be some leeway to relax this threshold requirement, otherwise a fair number of potentially eligible patients might be missed.

We agree with this comment; it is very difficult to predict this and shall discuss within our trial steering group whether we should reduce this period to 48 hours in order to ensure eligible patients are not missed as a result of the 72 hour stipulation.

c. Hyperoxemia has been shown to be independently associated with an increased in-hospital mortality among patients who are admitted after cardiac arrest ^{8,9}. It is probably worth including these patients too.

We agree that hyperoxaemia is to be avoided in those patients who have been admitted post cardiac arrest. However, we felt uncomfortable randomising patients with a potential brain injury into an intervention arm of permissive hypoxaemia. We felt that in the absence of safety data from prospective studies for this cohort, we should exclude post-cardiac arrest patients as targeting their SpO₂ at 88-92% might be disadvantageous for them.

d. Likewise, hyperoxemia is also shown to be independently associated with higher in-hospital mortality among ventilated patients admitted with traumatic brain injury ¹⁰. The brain trauma foundation guideline recommends avoiding hypoxia (PaO₂ <60 mmHg or SaO₂ <90%) among patients with neurotrauma ¹¹. These thresholds are well within the intervention range of this pilot RCT.

Would it be worth including all such patients and pre-specify these subgroups analyses in the larger definitive trial?

As per our comments regarding post cardiac arrest patients, we decided that we could not include patients with a traumatic brain injury as robust evidence supporting the safety of targeting their SpO₂ at 88-92% is absent. Furthermore, in a small feasibility study such as this, sub-group analysis will be statistically challenging. Lastly, neither of the ICUs involved in this study receives patients with traumatic brain injuries as they are both general ICUs (not neurosurgical).

2. I am intrigued to learn that a SpO₂ target range of 98-100% can be considered as 'standard care' at NHS hospitals. I think over last few years the standard care in relation to oxygenation has certainly shifted towards avoiding use of too much oxygen. The concern with using a default SpO₂ target of 96-100% as 'control' is that it may end up being somewhat dissimilar to the actual standard care by the time the trial is completed. For example, not many contemporary intensivists would be comfortable aiming for SpO₂ >96% while FiO₂ is >0.60 or >0.70 in a ventilated patient with respiratory failure, particularly when all the ARDS RCTs targeted a SpO₂ range of 88-95% ¹²⁻¹⁴. Ventilating at a FiO₂ of 1 is demonstrably harmful as recently shown in a large multicenter RCT ¹⁵. Those surveys that have been referenced seem outdated in context of several recent publications highlighting the potential harms of hyperoxia ¹⁵⁻¹⁸. This is also reflected in the 2017 British Thoracic Society recommendations to aim for a SpO₂ target range of 94-98% among critically ill patients ¹⁹; or in other words, to avoid SpO₂ >98%, the usual threshold for hyperoxia. I understand it is important to achieve adequate separation in the overall cohort, but this perhaps needs to be balanced against the risk of having a somewhat "artificial" standard care arm. It might be worth considering either having a true standard care arm, in which clinicians are free to prescribe oxygen as they would normally do in their day-to-day practice, or limit the FiO₂ titration range for the purpose of this study to 0.21-0.80 as done in a previous pilot RCT ²⁰.

Thank you for raising the pertinent and somewhat controversial topic of control group (standard therapy) management. We did, however, wonder why the reviewer assumed the standard of care in NHS hospitals is an SpO₂ of 98-100% as this is not mentioned anywhere in our manuscript and the target SpO₂ in our control group is 96% or above. As far as we are aware there are no large data sets from the UK to confirm what 'standard' practice is but the authors all agree that a threshold of 96% is likely to reflect common practice in the UK.

Helmerhorst et al. reported a mean PaO₂ of 12.9 kPa in a large sample of Dutch Critical Care Blood Gas results in 2014 (*Annals of Intensive Care* 2014, 4:23). This approximately corresponds to an SpO₂ in the high 90s and probably reflects UK practice too.

Many RCTs have been designed less than adequately as a result of a poorly thought out control group, or unrealistic 'standard' practice being advocated as a comparator. Our study team put a considerable amount of thought into this matter during the design of the study. We are unaware of any standard practice or uniform guidelines that applies to the oxygenation of critically ill patients on a mechanical ventilator in the NHS. There is advice regarding the management of patients with ARDS, but these patients form a relatively small proportion of those admitted to ICUs in the UK - recently reported as 12.5% (1). Even for this group, no evidence-based formal guidelines exist regarding oxygenation; as highlighted to one of the other reviewers, the recent ATS/ESICM guidelines for the treatment of patients with ARDS made no reference to optimum oxygenation in these patients (2). Oxygenation practice in mechanically ventilated patients in the UK is therefore (and understandably) highly varied. It is for this reason that we took the matter to an annual meeting of the UK Critical Care Research Forum and asked in an open debate if clinicians would be prepared to administer both interventions (permissive hypoxaemia and control) to patients given the lack of evidence supporting either approach. The majority of clinicians agreed this would be the most sensible solution to a complex issue, and one that was ethically sound. Our concern with allowing clinicians to oxygenate patients at their own discretion in a control group was that this would lead to enormous heterogeneity, even on a day to day basis, making interpretation of results extremely challenging. One further factor to consider, is that this field is in a continual state of flux. In the space of only a few years the landscape in this arena has changed dramatically with the publication of a number of small studies, however, no large-scale prospective multi-centre trial data exists to demonstrate benefit or harm in either of the interventional arms that we have selected for this study. As such we are confident that clinicians will engage with the administration of the control arm, and that it represents a reasonable reflection of current practice in the UK as a whole.

We have added text to the introduction to highlight that determining the validity of our control group is an important part of this feasibility study.

1) Summers C et al. Incidence and recognition of acute respiratory distress syndrome in a UK intensive care unit. *Thorax*. 2016 Nov; 71(11): 1050–1051.

2) Fan E et al. An Official American Thoracic Society/European Society of Intensive Care Medicine/Society of Critical Care Medicine Clinical Practice Guideline: Mechanical Ventilation in Adult Patients with Acute Respiratory Distress Syndrome. *Am J Respir Crit Care Med*. 2017 May 1;195(9):1253-1263.

3. It would be ideal to further minimize the pre-randomization duration of mechanical ventilation. It appears that the intention is to keep it under 24 hours, but perhaps keeping it under 12 hours would go a fair way in ascertaining that patients assigned to the 88-92% SpO₂ target would not be exposed to standard liberal oxygenation for prolonged periods prior to randomization. This would minimize "contamination".

We wholeheartedly agree with the aim of reducing the pre-randomisation time, however, due to the practicalities of screening, approaching families, and consent we felt that 24 hours was a reasonable compromise. At both sites we aim to recruit in as timely a manner possible and reducing the window from 24 to 12 hours would dramatically reduce our number of eligible patients.

4. The rationale to limit exposure to high SpO₂, that might occur in the standard practice, is also applicable to the period after extubation. Therefore, it may be questioned why the study intervention should cease following the end of invasive mechanical ventilation. This could run a risk of exposing patients assigned to 88-92% SpO₂ arm to liberal oxygen targets after extubation, potentially diluting the 'intervention effect', if there is any. The use of high flow oxygen through nasal cannula is increasing in ICUs. In my view, the study intervention should perhaps continue until the patient is liberated from all respiratory support, rather than just mechanical ventilation.

This is a very pertinent fact and one we considered at length in the design of the study. However, the primary purpose of this study is not to determine the efficacy of the intervention, but the feasibility of the study design. There having been no studies of this nature in the UK, it was our goal to assess the ability to recruit patients into a study of this nature and for clinicians to be comfortable managing patients in both interventional arms. We also felt that continuing the study beyond extubation may lead to extremely long intervention periods for some long-stay ICU patients that may move the focus of the study away from our intended goals. For future studies we agree that this would be a very important component of patient management to study.

4. The accuracy of pulse oximetry depends on the perfusion adequacy or the pulse amplitude of the selected monitoring site. Since the FiO₂ in this trial will be titrated to the assigned SpO₂ range measured by pulse oximetry, the investigators do not describe whether, or how, perfusion adequacy or the pulse amplitude will be checked amongst participants. Would health care professionals, who are caring for participants, try different monitoring sites to check where the best pulse amplitude is obtained, and/or check SpO₂ against CO-oximetry measured SaO₂ to ensure the accuracy of the targeted parameter? For example,

a. Use of restraints in ICU is not that uncommon. It is an established factor that may result in an inherent inaccuracy if pulse oximetry is performed in a restrained limb.

The use of patient restraints in the UK is exceedingly rare, if not unheard of, so this will not be a relevant factor in the recruiting centres (both are in the UK).

b. Different fingers may have different perfusion indices.

We'll ask bedside nurses to look at plethysmography trace and add to our guidelines that this should be considered if signals appear substandard.

c. An indwelling line in the supplying artery may potentially interfere with the perfusion index. Will the pulse oximeter probes exclusively placed on the limb without an invasive arterial cannula?

We'll add to guidelines that this should also be considered if plethysmography signals are considered to be substandard.

5. In addition, pulse oximeters can be of various types. Some use transmission method, some use reflectance, some are clip-on types, whereas others are wrap-arounds or wear-on rubber types. These may all have different biases and agreements limits when compared to a true SaO₂. Will the investigators mandate the use of single type pulse oximeters for all participants? Or will this data be also collected and presented for the participants?

This is indeed true but we have no intention of standardising the oximeters used as our intention was to try and keep the study as 'real world' and straight forward as possible. Whilst this certainly can be an issue, we feel that if clinical decisions are already being made on the basis of the bedside monitoring, it must be robust enough to facilitate a study of this nature. Standardising oximeters throughout and ICU is not routine practice in the UK.

6. It is not that clear from the manuscript what specific tests for oxidative stress will be performed. Are you considering any global parameter for oxidative stress such as the plasma redox potential based on CyS-CySS ratio, derived from the Nernst equation ($E_h \text{ CyS/ CySS} = -250 + 30 \log \frac{[\text{CyS}]}{[\text{CySS}]^2}$)²¹⁻²³.

Our oxidative stress marker plan is as follows:

- 4-Hydroxynonenal (4-HNE)
- Protein carbonyls
- Total antioxidant capacity
- Glutathione reductase

These markers are already listed in the article. We shall not be calculating the plasma redox potential.

Minor:

7. It is well known that the effects of O₂ on the lungs and brain are considered as part of a patho-physiological continuum in which a common factor is Nitric oxide (NO) and its interaction with other reactive species²⁴ that can produce a toxic oxidant called peroxynitrite, which may result in a selective nitration of tyrosine in proteins to create nitrotyrosine²⁵. Since the occurrence of protein tyrosine nitration under disease conditions is an established marker of NO-dependent oxidative stress²⁶, will the investigators consider measuring the plasma levels of protein bound 3-nitrotyrosine levels in this study?

Unfortunately, we have no funding to add further analysis of blood samples. Should additional funding be obtained we shall certainly consider this as an option.

8. Will the assigned SpO₂ targets remain in place at times of hemodynamic instability or during trips outside ICU for therapeutic or diagnostic purposes?

During time off of the ICU for interventions / investigations, the protocol will be paused, and then resumed on the patient's return. We have clarified this in the article. If a patient becomes haemodynamically unstable to the point where clinicians or the research team have concerns, the patient will be withdrawn from the study if it is felt to be in their best interest. This is an important component in assessing the feasibility of the study and we have added text to the article to express this.

9. It seems unclear if the primary and secondary outcome measures will be presented for each arm separately. If this is not planned, that is understandable for a feasibility study. However, it would be important to show whether there was a meaningful separation in the achieved SpO₂, SaO₂, PaO₂, and FiO₂. Is this part of the planned analysis?

As this is a feasibility study, no formal comparative analyses are planned. The primary and secondary outcome measures will be presented using summary statistics (e.g. means, standard deviations, medians, proportions). Safety data will be presented for each arm separately (see reply 2 to reviewer 2).

10. If the comparative analysis is not planned, then will the patients enrolled in this pilot-RCT be included in the larger RCT later on?

No, this is a feasibility study (not a pilot study) so the results will not be incorporated into a future larger trial.

11. Perhaps the study posters for both arms should clearly emphasize that "the lowest possible FiO₂ should be used to achieve the target SpO₂ range". This would be helpful in achieving best possible separation.

The guidelines for the permissive hypoxaemia group already contain the statement "Aim to use the lowest FIO₂ possible to achieve the target SpO₂." – see supplementary material document 1. We chose not to include this statement in the control group guidelines as this is not standard practice in the UK.

References

1. Esteban A, Anzueto A, Frutos F, et al. Characteristics and outcomes in adult patients receiving mechanical ventilation: a 28-day international study. *Jama*. 2002;287(3):345-355.
2. Brun-Buisson C, Minelli C, Bertolini G, et al. Epidemiology and outcome of acute lung injury in European intensive care units. Results from the ALIVE study. *Intensive Care Med*. 2004;30(1):51-61.

3. Aggarwal NR, D'Alessio FR, Tsushima K, et al. Moderate oxygen augments lipopolysaccharide-induced lung injury in mice. *American journal of physiology. Lung cellular and molecular physiology*. 2010;298(3):L371-381.
4. Altemeier WA, Sinclair SE. Hyperoxia in the intensive care unit: why more is not always better. *Curr Opin Crit Care*. 2007;13(1):73-78.
5. Knight PR, Kurek C, Davidson BA, et al. Acid aspiration increases sensitivity to increased ambient oxygen concentrations. *Am J Physiol Lung Cell Mol Physiol*. 2000;278(6):L1240-1247.
6. Matthay MA, Zimmerman GA. Acute lung injury and the acute respiratory distress syndrome: four decades of inquiry into pathogenesis and rational management. *Am J Respir Cell Mol Biol*. 2005;33(4):319-327.
7. Thiel M, Chouker A, Ohta A, et al. Oxygenation inhibits the physiological tissue-protecting mechanism and thereby exacerbates acute inflammatory lung injury. *PLoS Biol*. 2005;3(6):e174.
8. Kilgannon JH, Jones AE, Shapiro NI, et al. Association between arterial hyperoxia following resuscitation from cardiac arrest and in-hospital mortality. *Jama*. 2010;303(21):2165-2171.
9. Kilgannon JH, Jones AE, Parrillo JE, et al. Relationship between supranormal oxygen tension and outcome after resuscitation from cardiac arrest. *Circulation*. 2011;123(23):2717-2722.
10. Rincon F, Kang J, Vibbert M, Urtecho J, Athar MK, Jallo J. Significance of arterial hyperoxia and relationship with case fatality in traumatic brain injury: a multicentre cohort study. *Journal of neurology, neurosurgery, and psychiatry*. 2014;85(7):799-805.
11. The Brain Trauma Foundation. The American Association of Neurological Surgeons. The Joint Section on Neurotrauma and Critical Care. Resuscitation of blood pressure and oxygenation. *J Neurotrauma*. 2000;17(6-7):471-478.
12. Brower RG, Lanken PN, MacIntyre N, et al. Higher versus lower positive end-expiratory pressures in patients with the acute respiratory distress syndrome. *N Engl J Med*. 2004;351(4):327-336.
13. Wiedemann HP, Wheeler AP, Bernard GR, et al. Comparison of two fluid-management strategies in acute lung injury. *N Engl J Med*. 2006;354(24):2564-2575.
14. Acute Respiratory Distress Syndrome N, Brower RG, Matthay MA, et al. Ventilation with lower tidal volumes as compared with traditional tidal volumes for acute lung injury and the acute respiratory distress syndrome. *N Engl J Med*. 2000;342(18):1301-1308.
15. Asfar P, Schortgen F, Boisrame-Helms J, et al. Hyperoxia and hypertonic saline in patients with septic shock (HYPER2S): a two-by-two factorial, multicentre, randomised, clinical trial. *Lancet Respir Med*. 2017;5(3):180-190.
16. Stub D, Smith K, Bernard S, et al. Air Versus Oxygen in ST-Segment-Elevation Myocardial Infarction. *Circulation*. 2015;131(24):2143-2150.
17. Damiani E, Adrario E, Girardis M, et al. Arterial hyperoxia and mortality in critically ill patients: a systematic review and meta-analysis. *Crit Care*. 2014;18(6):711.
18. Helmerhorst HJ, Roos-Blom MJ, van Westerloo DJ, de Jonge E. Association Between Arterial Hyperoxia and Outcome in Subsets of Critical Illness: A Systematic Review, Metaanalysis, and Meta-Regression of Cohort Studies. *Crit Care Med*. 2015.
19. O'Driscoll BR, Howard LS, Earis J, Mak V. British Thoracic Society Guideline for oxygen use in adults in healthcare and emergency settings. *BMJ Open Respir Res*. 2017;4(1):e000170.
20. Panwar R, Hardie M, Bellomo R, et al. Conservative versus Liberal Oxygenation Targets for Mechanically Ventilated Patients. A Pilot Multicenter Randomized Controlled Trial. *Am J Respir Crit Care Med*. 2016;193(1):43-51.
21. Jones DP, Go YM, Anderson CL, Ziegler TR, Kinkade JM, Jr., Kirlin WG. Cysteine/cystine couple is a newly recognized node in the circuitry for biologic redox signaling and control. *FASEB J*. 2004;18(11):1246-1248.
22. Jones DP, Carlson JL, Mody VC, Cai J, Lynn MJ, Sternberg P. Redox state of glutathione in human plasma. *Free radical biology & medicine*. 2000;28(4):625-635.
23. Netto LE, de Oliveira MA, Monteiro G, et al. Reactive cysteine in proteins: protein folding, antioxidant defense, redox signaling and more. *Comp Biochem Physiol C Toxicol Pharmacol*.

2007;146(1-2):180-193.

24. Allen BW, Demchenko IT, Piantadosi CA. Two faces of nitric oxide: implications for cellular mechanisms of oxygen toxicity. *Journal of applied physiology*. 2009;106(2):662-667.

25. Beckman JS, Koppenol WH. Nitric oxide, superoxide, and peroxynitrite: the good, the bad, and ugly. *The American journal of physiology*. 1996;271(5 Pt 1):C1424-1437.

26. Radi R. Nitric oxide, oxidants, and protein tyrosine nitration. *Proc Natl Acad Sci U S A*. 2004;101(12):4003-4008.

VERSION 2 – REVIEW

REVIEWER	Huaiwu,He PUMCH
REVIEW RETURNED	16-Jun-2018

GENERAL COMMENTS	The revised text has been improved. No more comment
---

REVIEWER	Satoshi Suzuki Okayama University Hospital, Okayama, Japan
REVIEW RETURNED	30-Jun-2018

GENERAL COMMENTS	The authors have sufficiently responded to my question/comment.
---

REVIEWER	Rakshit Panwar ICU staff specialist, John Hunter Hospital, School of Medicine and Public Health, The University of Newcastle, Australia
REVIEW RETURNED	15-Jun-2018

GENERAL COMMENTS	I thank the authors for addressing each of my queries. I appreciate it. I generally agree with their responses but I have some lingering concerns about the implications of this study, particularly for the definitive RCT that is likely to follow this study. Please see below: 1) My previous comment whether a SpO₂ range of 98-100% can be considered as part of 'standard care' was in reference to the statement on page 13, line 20 (section on 'comparator'), where target SpO₂ range of 96% or above (i.e., 96-100%) is referred to as 'standard care'. In light of recent studies (point 2 below), and the 2017 British Thoracic Society recommendations (aim for a SpO₂ target range of 94-98% among critically ill patients), I wonder if the SpO₂ target range for the comparator arm should be limited to 96-98% (and thus avoid deliberate hyperoxia) in such trials? The implications are really for the definitive RCT that you'd do after this feasibility study. An obvious confounder for interpretation of the trial results, particularly if the comparator arm is perceived to be different than the acceptable standard care, would be- whether a deliberate or a protocolized "disadvantage" of the comparator arm could have made the intervention arm look good in comparison? 2) Further, a very recent meta-analysis, (https://www.thelancet.com/journals/lancet/article/PIIS0140-6736(18)30479-3/fulltext), the IOTA study, concludes that supplemental oxygen might be unfavorable above an SpO₂ target of 96%. ARDS patients, although a relatively small proportion of ICU patients overall, are likely to form a significant proportion of
--

	patients enrolled in trials such as this. LUNGSAFE study demonstrates this well. About a quarter of patients requiring mechanical ventilation had ARDS in that study, and clinicians tend to under-recognize ARDS. In this regard, another recent study (https://www.ncbi.nlm.nih.gov/pubmed/29261565), based on ten major RCTs among ARDS patients, observed a dose-response relationship between the cumulative above goal oxygen exposure (FiO2 >0.50 while PaO2 >80 mmHg) and worse clinical outcomes for participants with any level of ARDS severity. Therefore, my concern is that accumulating evidence is stacking up against liberal oxygenation targets, which might impact on the "comfort" level of policymakers or the clinicians who are managing patients randomized to higher SpO2 targets, particularly when FiO2 requirement to achieve the set SpO2 target exceeds 0.70 or 0.80. In my view, avoiding hyperoxia (SpO2 >98%) is becoming or will soon become a part of standard care and may be allowing this in the protocol in some way or form may allay some of these concerns. 3) Regarding feasibility outcomes, it might be useful to assess some study-specific measure of separation in oxygenation between the two arms, such as either a pooled frequency histogram of the percentage of time spent at each SpO2 level, or PaO2/ FiO2/ SpO2 separation in both arms, as demonstrated in other similar feasibility trials (https://www.atsjournals.org/doi/abs/10.1164/rccm.201505-1019oc). The reason this is important is that the whole study protocol is geared towards achieving this separation between the two arms. Readers would certainly be interested in knowing whether an adequate separation was achieved or not in the end. 4) Regarding page 7, line 10, "...excess of adverse events in the low SpO2 group" was not the conclusion of that trial, which found no significant between-group differences in regard to any of the measures of organ dysfunction (delta SOFA score, delta PaO2/FiO2, new-onset ARDS, delta creatinine, hemodynamic instability, vasopressor-free days, arrhythmia-free days, or ventilator-free days), or ICU or 90-day mortality. Therefore, I'd suggest reverting back to the statement that the authors wrote originally.
--	---

VERSION 2 – AUTHOR RESPONSE

Reviewers' Comments to Author:

Reviewer: 1

Reviewer Name: Huaiwu,He

Institution and Country: PUMCH, China

Please state any competing interests or state 'None declared': None declared

The revised text has been improved. No more comment

Reviewer: 2

Reviewer Name: Satoshi Suzuki

Institution and Country: Okayama University Hospital, Okayama, Japan

Please state any competing interests or state 'None declared': None declared

The authors have sufficiently responded to my question/comment.

Thank you – we appreciate the time and energy that reviewer 1 and 2 have put into the peer review process and the improvements in the manuscript that have resulted.

Reviewer: 3

Reviewer Name: Rakshit Panwar

Institution and Country: ICU staff specialist, John Hunter Hospital, School of Medicine and Public Health, The University of Newcastle, Australia

Please state any competing interests or state 'None declared': None declared

We disagree with the statement above (please see letter to Editor)

I thank the authors for addressing each of my queries. I appreciate it. I generally agree with their responses but I have some lingering concerns about the implications of this study, particularly for the definitive RCT that is likely to follow this study. Please see below:

1) My previous comment whether a SpO₂ range of 98-100% can be considered as part of 'standard care' was in reference to the statement on page 13, line 20 (section on 'comparator'), where target SpO₂ range of 96% or above (i.e., 96-100%) is referred to as 'standard care'. In light of recent studies (point 2 below), and the 2017 British Thoracic Society recommendations (aim for a SpO₂ target range of 94-98% among critically ill patients), I wonder if the SpO₂ target range for the comparator arm should be limited to 96-98% (and thus avoid deliberate hyperoxia) in such trials? The implications are really for the definitive RCT that you'd do after this feasibility study. An obvious confounder for interpretation of the trial results, particularly if the comparator arm is perceived to be different than the acceptable standard care, would be- whether a deliberate or a protocolized "disadvantage" of the comparator arm could have made the intervention arm look good in comparison?

Many thanks to the reviewer for highlighting this interesting and controversial component of studies that aim to determine optimum oxygenation in the critically ill. The BTS guideline that is referred to is very familiar to us. Dr Ronan O'Driscoll, an investigator in our team, is an author on these guidelines and led the team that developed them and wrote the manuscript. The guidance provided in the document is for "acutely unwell" patients and it gives no recommendations for optimal oxygenation levels in patients receiving mechanical ventilation. Specifically, "oxygen use in ICUs" is listed as an "area not covered by this guideline". Whilst one could postulate that the optimum oxygenation in these two patient groups (acutely unwell / on ICU) may be similar, there is an equally valid argument that they may be different (in either direction). We therefore feel that our standard of care group does not fall outside of what is currently practiced in the UK. We are aware that other trials have used what we would regard as unnecessarily high levels of oxygenation in their "standard of care" group (e.g. Girardis et al. 2016) and we have done everything possible to avoid this. We note that in the protocol for the recently completed ICU-ROX trial (of which we believe this reviewer is an investigator), the 'standard' group had "no specific measures taken to limit FiO₂ or SpO₂ and the use of upper alarm limits for SpO₂ will be prohibited". Furthermore, it is stated within the ICU-ROX trial protocol that, "To minimise the risk of contamination, the use of FiO₂ < 0.3 while patients are invasively ventilated will be discouraged." We chose not to implement a minimum FiO₂, precisely to avoid unnecessary hyperoxaemia. The selection of targets and regimens for standard of care groups in these types of trials will remain challenging in this constantly changing landscape.

To support our choice of oxygenation target in the standard of care group in our trial we refer the reviewer to the recent publication by Schjørring et al (2018). In summary, more than three quarters of 1080 European CCU doctors (including 202 UK doctors) would accept a PaO₂ target of 12 kPa or higher in a clinical trial of oxygenation targets. This translates to an approximate SpO₂ of 97% or above as per table 10 of the BTS Guidelines mentioned above (O'Driscoll et al 2015). Within this group, a minority of doctors would abide to an upper oxygenation limit of 12kPa and the majority were happy to go higher than 12 kPa (i.e. to accept an upper SpO₂ limit of ≥ 97%). Thus, there is clear proof of equipoise amongst European Critical Care specialists (the setting for our study) for the target we propose (SpO₂ upper limit ≥96% saturation).

Schjørring OL et al. Intensive care doctors' preferences for arterial oxygen tension levels in mechanically ventilated patients. *Acta Anaesthesiol Scand*. 2018

Lastly, we would like to draw the reviewer's attention to the advice provided for peer reviewers of study protocols on the BMJ Open website:

"Reviewers will be instructed to review for clarity and sufficient detail. The intention of peer review is not to alter the study design. Reviewers will be instructed to check that the study is scientifically credible and ethically sound in its scope and methods, and that there is sufficient detail to instil confidence that the study will be conducted and analysed properly."

<https://bmjopen.bmj.com/pages/authors/>

2) Further, a very recent meta-analysis, ([https://www.thelancet.com/journals/lancet/article/PIIS0140-6736\(18\)30479-3/fulltext](https://www.thelancet.com/journals/lancet/article/PIIS0140-6736(18)30479-3/fulltext)), the IOTA study, concludes that supplemental oxygen might be unfavorable

above an SpO₂ target of 96%. ARDS patients, although a relatively small proportion of ICU patients overall, are likely to form a significant proportion of patients enrolled in trials such as this. LUNGSAFE study demonstrates this well. About a quarter of patients requiring mechanical ventilation had ARDS in that study, and clinicians tend to under-recognize ARDS. In this regard, another recent study (<https://www.ncbi.nlm.nih.gov/pubmed/29261565>), based on ten major RCTs among ARDS patients, observed a dose-response relationship between the cumulative above goal oxygen exposure (FiO₂ >0.50 while PaO₂ >80 mmHg) and worse clinical outcomes for participants with any level of ARDS severity. Therefore, my concern is that accumulating evidence is stacking up against liberal oxygenation targets, which might impact on the "comfort" level of policymakers or the clinicians who are managing patients randomized to higher SpO₂ targets, particularly when FiO₂ requirement to achieve the set SpO₂ target exceeds 0.70 or 0.80. In my view, avoiding hyperoxia (SpO₂ >98%) is becoming or will soon become a part of standard care and may be allowing this in the protocol in some way or form may allay some of these concerns.

We are also very familiar with the IOTA paper, which is a systematic review and meta-analysis, rather than a study. Again, this article relates to acutely unwell patients, not specifically to those patients in whom mechanical ventilation has been initiated on an ICU. This analysis has a number of serious issues, which are outlined in our accepted letter to the editor of the journal (in press). To summarise here:

1. The heterogeneity of disease categories included in the analysis was substantial, including patients with stroke, trauma, sepsis and those who had undergone emergency surgery.
2. The range of interventions used in these studies was similarly broad with some studies comparing strictly administered oxygen concentrations whilst others compared tightly defined arterial oxygenation targets.
3. As the reviewer has raised above, liberal / standard oxygen therapy in these studies often involved the administration of oxygen at levels well outside normal clinical practice.
4. Two of the studies (one in myocardial infarction and one in stroke) contributed more than 70% of the included patients (and more than 1/3 of deaths).

We therefore feel that whilst the mathematical outcomes of this meta-analysis might suggest harm from liberal oxygen therapy in acutely unwell patients, attempting to translate this into the setting of mechanically ventilated critically ill patients is at best questionable and at worst dangerous.

In the UK the incidence of ARDS amongst mechanically ventilated critically ill patients has been estimated at 12.5 % (Summers et al.) and broadly speaking this would represent the population from which we would recruit to this and any future studies.

Finally, whether our lower boundary for the standard of care group is 94% or 96% or the higher boundary is 98% or 100% is in both instances based on a difference in SpO₂ of 2%. If a difference in SpO₂ of only 2% was able to exert a substantial difference on clinical outcomes, we think this discussion would be of great value. However, given the limited accuracy of most peripheral saturation probes we are not convinced it requires further discussion.

3) Regarding feasibility outcomes, it might be useful to assess some study-specific measure of separation in oxygenation between the two arms, such as either a pooled frequency histogram of the percentage of time spent at each SpO2 level, or PaO2/ FiO2/ SpO2 separation in both arms, as demonstrated in other similar feasibility trials (<https://www.atsjournals.org/doi/abs/10.1164/rccm.201505-1019oc>). The reason this is important is that the whole study protocol is geared towards achieving this separation between the two arms. Readers would certainly be interested in knowing whether an adequate separation was achieved or not in the end.

Many thanks to reviewer 3 for referencing his paper on this topic. We are happy to accept this suggestion as a useful illustrative way of presenting the trial data.

4) Regarding page 7, line 10, "...excess of adverse events in the low SpO2 group" was not the conclusion of that trial, which found no significant between-group differences in regard to any of the measures of organ dysfunction (delta SOFA score, delta PaO2/FiO2, new-onset ARDS, delta creatinine, hemodynamic instability, vasopressor-free days, arrhythmia-free days, or ventilator-free days), or ICU or 90-day mortality. Therefore, I'd suggest reverting back to the statement that the authors wrote originally.

Many thanks to the reviewer for highlighting this. This was a typographical error in which the word "no" had been omitted! We have amended this sentence.

VERSION 3 – REVIEW

REVIEWER	Rakshit Panwar John Hunter Hospital, Australia
REVIEW RETURNED	17-Oct-2018
GENERAL COMMENTS	I thank the authors for addressing each of my points. I appreciate it. As far as the ANZICS trial is considered, the trial is long finished in terms of recruitment. If people who have conducted similar previous trials stop reviewing similar relevant papers in future, then I think it'd be a disservice to scientific rigor involved in the peer review process. The issues that I highlighted in my review of your paper arose from the practical concerns expressed by the clinicians looking after patients that were participating in our previous RCT. I was only attempting to see if what we couldn't incorporate, or got too late to address, could be addressed in other subsequent RCTs. I did not realise that you have acquired all funding and ethics approval for the main RCT even before the feasibility trial is completed. You listed under the 'strengths of your study' that "this study will provide valuable information to enable the design of future large-scale RCT". Anyways, as far as the duration of peer review is considered, my previous reviews were well within the recommended time period for review as stipulated by the journal guidelines. This last review got slightly delayed as I am on overseas vacation and did not check my emails. So my apologies if you believe I have delayed the acceptance of your

	paper. Best wishes for your definitive RCT. I will certainly look forward to the findings.
--	--